# PROSPECT-PMP+: Simultaneous Retrievals of Chlorophyll a and b, Carotenoids and Anthocyanins in the Leaf Optical Properties Model

**DOI:** 10.3390/s22083025

**Published:** 2022-04-14

**Authors:** Yao Zhang, Xinkai Li, Chengjie Wang, Rongxu Zhang, Lisong Jin, Zongtai He, Shoupeng Tian, Kaihua Wu, Fumin Wang

**Affiliations:** 1College of Artificial Intelligence, Hangzhou Dianzi University, Hangzhou 310018, China; zhangyao@hdu.edu.cn (Y.Z.); lxk@hdu.edu.cn (X.L.); wangchengjie@hdu.edu.cn (C.W.); zrx@hdu.edu.cn (R.Z.); jls@hdu.edu.cn (L.J.); hzt1997@hdu.edu.cn (Z.H.); tianshoupeng@hdu.edu (S.T.); 2Institute of Agricultural Remote Sensing & Information Application, Zhejiang University, Hangzhou 310058, China; wfm@zju.edu.cn; 3Key Laboratory of Agricultural Remote Sensing and Information System, Hangzhou 310058, China

**Keywords:** photosynthetic and photo-protective pigments, LOPEX_ZJU, multiple pigment absorption feature separation, PROSPECT-MP+

## Abstract

The PROSPECT leaf optical radiative transfer models, including PROSPECT-MP, have addressed the contributions of multiple photosynthetic pigments (chlorophyll a and b, and carotenoids) to leaf optical properties, but photo-protective pigment (anthocyanins), another important indicator of vegetation physiological and ecological functions, has not been simultaneously combined within a leaf optical model. Here, we present a new calibration and validation of PROSPECT-MP+ that separates the contributions of multiple photosynthetic and photo-protective pigments to leaf spectrum in the 400–800 nm range using a new empirical dataset that contains multiple photosynthetic and photo-protective pigments (LOPEX_ZJU dataset). We first provide multiple distinct in vivo individual photosynthetic and photo-protective pigment absorption coefficients and leaf average refractive index of the leaf interior using the LOPEX_ZJU dataset. Then, we evaluate the capabilities of PROSPECT-MP+ for forward modelling of leaf directional hemispherical reflectance and transmittance spectra and for retrieval of pigment concentrations by model inversion. The main result of this study is that the absorption coefficients of chlorophyll a and b, carotenoids, and anthocyanins display the physical principles of absorption spectra. Moreover, the validation result of this study demonstrates the potential of PROSPECT-MP+ for improving capabilities in remote sensing of leaf photosynthetic pigments (chlorophyll a and b, and carotenoids) and photo-protective pigment (anthocyanins).

## 1. Introduction

Leaf pigments contain multiple photosynthetic pigments (including chlorophyll a (Chla), b (Chlb), and carotenoids (Cars)) and photo-protective pigment (anthocyanins (Ants)) and are closely linked to vegetation physiological and ecological functions [1,2]. As Chla performs plant photosynthesis and Chlb assists Chla to perform photosynthesis in higher plant leaves, they have become the key measurement parameters in vegetation canopies [3]. β-carotenoid (β-Car) also transfers a faction of absorbed energy to Chla [4], and xanthophylls (including in violaxanthin (Vi), antheraxanthin (An), zeaxanthin (Ze)) protect the photosynthetic system by dissipating excess absorbed energy [5,6]. Ants also perform a protective function on Chla photosynthesis and indicate plant physiological and ecological status, especially at a lower temperature and with higher ultraviolet radiation [7,8,9]. The characteristics of leaf reflectance and transmittance spectra in the 400–800 nm region significantly depend on these pigment types and their content [10]. The development of precision monitoring for vegetation physiological and ecological status has also promoted the need for remote sensing of multiple plant pigments [11]. Thus, improved modelling of leaf optical properties can be achieved by better measurement and knowledge of the vegetation physiological and ecological characteristics [12]. However, to date, these photosynthetic and photo-protective pigments have not been simultaneously measured from remote sensing data, due to the overlapping of absorption of these pigments has not been considered.

The reflectance and transmittance of plant leaf mainly depend upon leaf biochemical (including leaf pigment groups) and biophysical characteristics [13,14,15]. Leaf radiative transfer models (RT models) can describe these processes and provide an opportunity for the accurate analysis of remotely sensed signals by quantifying the response of leaf electromagnetic radiation to pigment concentrations [16,17]. Various researchers have developed RT models for broad leaves that can run in forward mode for simulating leaf spectra and in backwards mode for retrieving pigment concentrations based on the given pigment absorption coefficients. Remarkably, PROSPECT [18] has become a key model to monitor the plant pigments by simulating leaf optical properties in the 400–800 nm region. Several versions of the PROSPECT model for the retrievals of different pigments in leaves have been released since 1990: they correspond to the separation of leaf Chls absorption coefficients [18], and of Chls and Cars absorption coefficients [19] in PROSPECT-5 with a minimum distance fitting of spectra; the adaptation of Fluspect-B [20] to account for leaf fluorescence, reflectance, and transmittance spectra fluorescence emission, and then the retrieval of Chls and Cars concentration; the development of PROSPECT-D [21] from a LOPEX dataset (ANGERS) with the computed photo-protective pigment (anthocyanins (Ants)) concentration, which then retrieves Chls, Cars, and Ants concentrations; the recent development of an algorithm for the determination of leaf multiple pigment absorption coefficients, enabling the masking phenomenon of different individual pigments in PROSPECT-MP to be limited [22], in the model for the retrievals of Chla, Chlb, and Cars (All the notations and units are specified in Table 1). To date, leaf photosynthetic pigments (Chla, Chlb, and Cars) and photo-protective pigment (Ants) are not simultaneously taken into account in this model, and are not simultaneously retrievable.

The monitoring of plant physiological and ecological status and pigment discrimination require finer and more pigments to be measured. In this study, a new dataset (LOPEX-ZJU) with Chla, Chlb, Cars, and Ants information was built, providing an opportunity for extending PROSPECT again. A new version for PROSPECT (PROSPECT-MP+) was developed based on a physically based description of pigment absorption coefficients. To test the ability of simultaneously separating multiple pigments in leaf spectra, The comparisons of spectral modelling and pigment inversion between PROSPECT-5, PROSPECT-D and PROSPECT-MP+ were performed using the new dataset that contains photosynthetic pigments (Chla, Chlb, and Cars) and photo-protective pigment (Ants).

## 2. Development of a New Leaf Optical Properties Experiment Data

The published leaf optical properties experiment data, such as LOPEX93, CALMIT, ANGERS, and HAWALL [19], do not contain photo-protective pigments (Ants) and only hold leaf photosynthetic (Chla, Chlb, and Cars) information. To extend PROSPECT function to simultaneously consider photosynthetic and photo-protective pigments, a new dataset was acquired specifically for this study at the Provincial Key Laboratory of Agricultural Remote Sensing and Information System in Zhejiang University, Hangzhou, China. This Leaf Optical Properties Experiment at ZheJiang University (LOPEX_ZJU) collected data that were equivalent to LOPEX93 but also enabled additional validation of the spectral modelling capabilities of PROSPECT-MP+ and the capacity to retrieve concentrations of leaf photo-protective pigments (Ants) by inversion.

The LOPEX_ZJU dataset was collected on Zhejiang University campus, which is located in the subtropical monsoon climate zone, from September to December in 2015. The leaf samples (Table 2) contained a range of 12 species with different biophysical characteristics, encompassing evergreen trees, deciduous trees, shrubs, subshrubs, and herbaceous plants. To obtain a wide range of variability in individual pigment concentrations, leaf samples from a range of different leaf growth stages, monitored using a SPAD meter [23], were obtained for each species, resulting in 59 leaf samples being collected for the study.

### 2.1. Leaf Radiometric Properties

Leaf directional hemispherical reflectance and transmittance (DHR and DHT) were measured by employing an integrating sphere attached to a spectrophotometer (UV-3600, Shimazdu) operating in the 240–2400 nm range. The instrument provides a spectral resolution of around 1 nm depending on the wavelength. The method for measuring the DHR and DHT of the leaf adaxial (upper) face follows the protocols of LOPEX93, DHT measurements being calibrated using a reflectance reference panel and the DHR measurements being corrected for the reflectance of the black background placed beneath the leaf samples using the following relationship [24,25]:(1)$$T_{\mathrm{mea}}\left(\lambda \right)=\frac{T_{\mathrm{sen}}{(\lambda )R}_{\mathrm{refer}}(\lambda )}{1-R_{\mathrm{bla}}(\lambda )};{\;R}_{\mathrm{mea}}\left(\lambda \right)=\frac{\left(R_{\mathrm{sen}}(\lambda )-R_{\mathrm{bla}}(\lambda )\right){\;R}_{\mathrm{refer}}(\lambda )}{1-R_{\mathrm{bla}}(\lambda )}$$
where $R_{\mathrm{sen}}(\lambda )$ and $T_{\mathrm{sen}}(\lambda )$ are the raw DHR and DHT with UV-3600 sensor, respectively; $R_{\mathrm{refer}}(\lambda )$ is the DHR of reference panel; $R_{\mathrm{bla}}(\lambda )$ is the DHR of the black background placed under the leaf sample; $R_{\mathrm{mea}}(\lambda )$ and $T_{\mathrm{mea}}(\lambda )$ are the calibrated DHR and DHT, which can be directly used for the parameter calibration or the performance evaluations of PROSPECT-MP+, PROSPECT-D, and PROSPECT-5.

### 2.2. Leaf Biophysical and Biochemical Properties

The extraction and separation of leaf photosynthetic pigments (Chla (chlorophyll a), Chlb (chlorophyll), β-Car (β-carotenoid), Vi (violaxanthin), An (antheraxanthin), Ze (zeaxanthin), Ne (Neoxanthin), and Lu (Lutein)) are difficult using a spectrophotometer with the frequently used wet chemical methods [18]. Instead, here we used the HPLC method, which has been shown to accurately determine photosynthetic pigment concentrations in fresh leaves [26,27]. The measurement campaign involved three steps: leaf pigment extraction, HPLC analysis, and leaf pigments content computation. In the first step, two leaf tissue disks (approximately 0.05 g) were bored from the fresh leaves using a cork borer (diameter = 0.97 cm), and pigments were extracted into solution following the method of Lee et al. [27]. In the second step, chromatography was carried out on a 4.6 × 150 mm Agilent C18 radial compression column (5 μm particle size). The extraction solutions were injected with an Agilent injector with a 20 μL loop, and mobile phases were pumped by Agilent 1200 high-pressure pump at a flow rate of 1 mL/min. The proportion and designated time of different mobile phases follow the work of De Las Rivas et al. [26]. In the final step, peaks were detected at 450 nm using a Shimadzu UV-V detector and integrated with a Shimadzu CR3 integrator, for the measurement of individual pigment concentrations [27].

To measure photo-protective pigment (Ants) content, two additional disks were collected from the same leaf sample used for photosynthetic pigments measurement. Ants were extracted in a solution of cold methanol/HCl/water (90:1:1 *v*/*v*/*v*) [28]. The absorbance values of the solution were measured with a UV-2550 spectrophotometer at 530 and 657 nm, respectively. To correct for the effect of chlorophyll on the Ants absorption at 530 nm, we used the empirical equation from Mancinelli [29] and Mancinelli and Schwartz [30]:(2)$$\mathrm{AA}=A_{530}-0.25A_{657}\;$$
where AA is a corrected value of Ants absorbance; A_530_ and A_657_ are absorbance of the Ants solution at 530 nm and 657 nm, respectively. A molar absorbance coefficient (30,000 g/mol/cm) [31] and AA were used to calculate leaf Ants content.

The concentration of each pigment within each leaf was expressed as a mass of pigment per unit area of leaf and calculated based on the known single-sided area of each leaf disk analyzed. In parallel with the spectral and pigment measurements, leaf water content was measured to obtain some auxiliary data to support the study, using the methods of Hosgood et al. [24].

## 3. Need for a Calibration for Leaf Absorption Coefficient for PROSPECT-MP+

The improved algorithm in PROSPECT-MP [22] was developed based on the Gauss–Lorentz function (GLF) to simultaneously separate leaf multiple photosynthetic pigment absorption coefficients with the overlapping characteristics. However, GLF fitting requires a known position of absorption peak. This problem in the improved algorithm was solved by determining the positions of single absorption peaks in organic solution and a small band shift parameter. In the calibration of PROSPECT model parameters, the band shift was designed in the small change range, which could limit the spectrum fitting in the absorption peak overlapping regions and prevent overfitting.

The parameters in the PROSPECT model were calibrated based on the spectral fitting of minimum distance. In the application of spectral fitting of minimum distance, the more parameters, the higher the risk of overfitting. However, compared with in PROPSECT-4, PORSPECT-5, and PROSPECT-D, the improved algorithm employing Gauss–Lorentz function for the description of leaf pigment absorption coefficients in PROSPECT-MP greatly reduced the number of calibrated parameters. For example, 401 parameters for the Cars (carotenoids) absorption coefficients in 400–800 nm were calibrated in PROSPECT-5, whereas only 12 parameters for Cars (carotenoids) absorption coefficients in 400–800 nm were calibrated in PROSPECT-MP. Thus, the improved algorithm employing Gauss–Lorentz function for the description of leaf pigment absorption coefficients could still avoid the risk of overfitting when leaf anthocyanins (Ants) were introduced in PROSPECT-MP.

According to the improved algorithm for calibrating leaf multiple pigment absorption coefficients in the PROSPECT-MP model [22], each absorption peak of absorption coefficients for leaf photosynthetic and photo-protective pigments in Equation (3a) is uniformly characterized by a modified Gauss–Lorentz function, and each pigment absorption coefficient is described in Equation (3b).
(3a)$$K_{i,j}\left(\lambda \right)=K_{i,j,v}\cdot K_{i,j,h}\cdot e^{-4\ln2\;\cdot \;{\left(\frac{A_{i,j,p}\;+\;K_{i,j,\Delta \lambda }-\lambda }{K_{i,j,w}}\right)}^{2}}+\left(1-K_{i,j,v}\right)\frac{K_{i,j,h}}{1+4{\left(A_{i,j,p}+K_{i,j,\Delta \lambda }-\lambda \right)}^{2}K_{i,j,w}{}^{-2}}$$
(3b)$$K_{i}\left(\lambda \right)=\overset{j}{\underset{j=1}{\sum }}K_{i,j}(\lambda )\;$$
where i is the calibrated pigment type (Chla, Chlb, Cars, and Ants); j is the peak number within the pigment-specific absorption coefficient (see Table 3); $K_{i,j}(\lambda )$ represents the jth peak function within the absorption coefficient for the ith pigment type; $K_{i,j,v}$, $K_{i,j,h}$, and $K_{i,j,w}$ are the Gauss ratio, peak height, and full width at half maximum (FWHM) of the jth absorption peak for the ith pigment type in vivo, respectively; $A_{i,j,p}$ is the peak position of the jth absorption peak for the ith pigment type in organic solution; $K_{i,j,\Delta (\lambda )}$ is the spectral displacement of the jth absorption peak for the ith pigment type in vivo. The factors i, *j*, and $A_{i,j,p}$ are given in Figure 1 and Table 4.

In order to simultaneously retrieve multiple photosynthetic and photo-protective pigments in a leaf optical PROSPECT model, we extend PROSPECT-MP to a new version (PROSPECT-MP+) for the simultaneous retrieval of Chla, Chlb, Cars, and Ants by modifying leaf absorption coefficients. According to the reports by Feret et al. [19] and Zhang et al. [22], the leaf absorption coefficient (k(λ)) incorporating Chla, Chlb, Cars, and Ants is described in the following equation:(4)$$k(\lambda )=\frac{K_{\mathrm{Chla}}{(\lambda )C}_{\mathrm{Chla}}+K_{\mathrm{Chlb}}{(\lambda )C}_{\mathrm{Chlb}}+K_{\mathrm{Cars}}{(\lambda )C}_{\mathrm{Cars}}+K_{\mathrm{Ants}}{(\lambda )C}_{\mathrm{Ants}}}{N}+K_{0}(\lambda )\;$$
where $K_{\mathrm{Chla}}(\lambda )$, $K_{\mathrm{Chlb}}(\lambda )$, $K_{\mathrm{Cars}}(\lambda ),$ and $K_{\mathrm{Ants}}(\lambda )$ stand for Chla, Chlb, Cars, and Ants specific absorption coefficients, respectively; $C_{\mathrm{Chla}}$, $C_{\mathrm{Chlb}}$,${\;C}_{\mathrm{Cars}}$ and $C_{\mathrm{Ants}}$ stand for Chla, Chlb, Cars, and Ants concentrations in the corresponding fresh leaf; $K_{0}\left(\lambda \right)$ stands for the baseline absorption coefficient for the absorption characteristics of the non-pigment photosensitive material in in vivo leaf; N stands for leaf structure index.

As in the PROSPECT-MP version, we apply the characterized k(λ) of Equation (4) into the PROSPECT model for a leaf optical model for the simultaneous retrieval of multiple photosynthetic (Chla, Chlb and Cars) and photo-protective pigments (Ants): PROSPECT-MP+.

## 4. Calibration and Evaluation of PROSPECT-MP+

### 4.1. Calibration of PROSPECT-MP+

Compared to the PROSPECT-MP version, the PROSPECT-MP+ includes the more factor of leaf optical property (Ants). This changes the optical features of Chla, Chlb, and Cars absorption coefficients and the leaf average refractive index (${\bar{m}}_{\mathrm{la}}$) because of their overlapping band characteristic. Thus, $K_{i}(\lambda )$ and ${\bar{m}}_{\mathrm{la}}(\lambda )$ for PROSPECT-MP+ were calibrated by minimizing the merit function with a least squares optimization:(5)$$\chi \left({\bar{m}}_{\mathrm{la}}\left(\lambda \right),K_{i}\left(\lambda \right)\right)=\underset{m}{\sum }\overset{800}{\underset{\lambda =400}{\sum }}{\left(R_{\mathrm{mea}}\left(\lambda \right)-R_{\mathrm{mod}}\left(\lambda \right)\right)}^{2}+{\left(T_{\mathrm{mea}}\left(\lambda \right)-T_{\mathrm{mod}}\left(\lambda \right)\right)}^{2}$$
where *m* stands for leaf sample number of the selected data from the LOPEX_ZJU dataset (*m* = 31), and those leaf samples are named the calibration dataset. The left leaf samples were used for the model evaluations (see the Section 4.2; *n* = 28) and were named the validation dataset. The selection standard of leaf samples for the calibration dataset and the validation dataset was that both the range and averaged values of leaf pigment content of the two selected datasets can represent the range and average value of the corresponding pigment content in the LOPEX_ZJU dataset. $R_{\mathrm{mea}}$ and $T_{\mathrm{mea}}$, $R_{\mathrm{mod},0}$, and $T_{\mathrm{mod},0}$ are the measured reflectance and transmittance, and the modeled reflectance and transmittance of the selected leaf samples, respectively.

In addition, as the leaf structure index of each leaf sample is required in the calibration and evaluation of PROSPECT-MP+, we employed the algorithm of leaf structure index reported by Feret et al. [19].

### 4.2. Evaluating the Performance of PROSPECT-MP+

To evaluate the performance and stability of PROSPECT-MP+ in leaf spectral modelling and pigment retrieving, different comparisons are displayed based on various implementations of the PROSPECT-MP+, PROSPECT-D, and PROSPECT-5 versions using LOPEX_ZJU datasets. The details of each implementation for the different PROSPECT versions are provided in Table 5 for of the employed dataset type, the number of leaf samples used, the input and output variables, and the algorithm employed.

In the spectral modelling evaluation, the metrics used were RMSE (root mean square error), BIAS, SEC (standard error corrected), and CV (coefficient variability), and in the pigment retrieval evaluation, the metrics used were RMSE, BIAS, SEC, and CV (coefficient variability) [19].

## 5. Results and Discussion

### 5.1. Parameter Calibration

Using a new LOPEX_ZJU dataset, $K_{\mathrm{Chla}}$ and $K_{\mathrm{Chlb}}$ can be separated from $K_{\mathrm{Chls}}$ using PMP+, similarly to PMP not do from [21] PD and P5. And $K_{\mathrm{Ants}}$ can also be separated, similar to PD, not do from PMP and P5. The ${\bar{m}}_{\mathrm{la}}\left(\lambda \right)$ spectra in the three versions (P5, PMP, and PM P+) of the PROSPECT model have a similar undulating shape, consistent with the results of Paillotin et al., who demonstrated that the refractive index is related with thylakoid membrane pigmentation [33], and this is different from that of PD.

PMP+, like PMP, employs the modified Gauss–Lorentz function, and the determined pigment absorption coefficients ($K_{\mathrm{Chla}}$, $K_{\mathrm{Chlb}}$, $K_{\mathrm{Cars}}$, and $K_{\mathrm{Ants}}$) in PMP+ (1) are also all in accordance with the physical principles underpinning pigment absorption spectra, especially $K_{\mathrm{Cars}}$ (these physical features are presented in P5 and PD (Figure 1)); (2) can directly account for peak position variations in environment polarity between the organic solution and a leaf in vivo by using the spectral displacement parameter (Table 6); (3) can also quantify the absorption characteristic of the each pigment in vivo using the range of absorption feature (RAF) parameter (Table 6) [22]. In addition, there are the similar positions of the first peak and the fourth peak from $K_{\mathrm{Chla}}$ and the third peaks from $K_{\mathrm{Chlb}}$ between the PMP and PMP+ (see Figure 2, Table 6, and Zhang et al. [22]). Those absorption peak positions and the range of absorption feature (RAF) of leaf pigment absorption coefficients are expected to be applied to the development of a professional sensor for plant pigment determining and monitoring.

However, there are some differences in the determined pigment-specific absorption coefficients ($K_{\mathrm{Chla}}$, $K_{\mathrm{Chlb}}$, and $K_{\mathrm{Cars}}$) between PMP and PMP+. The main absorption regions of $K_{\mathrm{Chla}}$, $K_{\mathrm{Chlb}}$, and $K_{\mathrm{Cars}}$ from PMP+ are higher than those from PMP. This is possibly due to the following differences: (1) PMP+ combined the effect of Ants, but this action was not performed in PMP, in which the Ants absorption feature overlapping other pigments was successfully separated in PMP+ and transferred to other pigments in the separation of pigment absorption coefficients in PMP. (2) Pigment concentration measurements in LOPEX_ZJU employed HPLC, which can precisely determine leaf pigment content, and LOPEX93 or ANGERS employed a spectrophotometry method that underestimated leaf Cars content [18]. For the absorption peak positions, there are visible differences in the first absorption peak of $K_{\mathrm{Chlb}}$ and $K_{\mathrm{Cars}}$ between PMP and PMP+, which is a result of the consideration, or not, of Ants in the two versions.

### 5.2. Performance Evaluation

In this section, we evaluate the spectral modelling and pigment retrieval capabilities of PROSPECT-MP+ based on the comparison with PROSPECT-5 and PROSPECT-D using the LOPEX_ZJU dataset. As it was demonstrated previously that the performance of PMP incorporating K0 is excellent [22], PROSPECT-5 also considered the effect of K0 on the spectral modelling and pigment retrieval capabilities. For PROSPECT-D version, the spectral modelling and pigment retrieval methods were following the report by Feret et al. [21].

#### 5.2.1. Spectral Modelling Performance

Figure 3 shows simulated and measured DHR and DHT spectra for three leaves of low, medium, and high Ants concentrations from the validation dataset. The performance of PMP+ is particularly effective for the low-Ants-concentration leaves, and it is encouraging for the medium-Ants-concentration leaves, with some overestimation in the 580–680 nm region. For the high-Ants-concentration leaves, there is some underestimation of DHR around 500–580 nm and overestimation at 650–700 nm, while the DHT simulation matches well with the measured spectrum. These performances of PD are similar to those of PMP+, except for in the 580–680 nm region. However, the spectral modelling performance of P5 is weaker than that of PMP+ and PD across all three different Ants concentrations. This can be attributed to the absence of an Ants absorption coefficient within the τ parameter in P5 [19].

#### 5.2.2. Spectral Modelling Evaluation

##### Global Performance Evaluation of Simulated Leaf Spectra

Based on validation against the LOPEX_ZJU dataset, the global performance of PMP+ and PD for leaf DHT and DHT modelling is excellent, as RMSE and SEC are both less than 0.03 and BIAS is lower than ±0.01 (see Table 7). P5 scores lower than PMP+ does in every evaluation metric. The results indicate that PMP+ has a superior capability for leaf spectra modelling, and P5 is much less effective. These results confirm that PMP+ can successfully simulate leaf spectra by incorporating Ants information while P5 lacks this ability.

##### Local Performance Evaluation of Simulated Leaf Spectra

In considering the local performances in spectral modelling, the largest errors generated by PMP+, PD, and P5 for DHR and DHT simulations, especially the RMSE and SEC metrics, are located in the 500–600 nm region (Figure 4). With respect to P5, this is because, as alluded to above, PROSPECT-5 does not incorporate an Ants absorption coefficient and the RAF of this pigment group is located in the 500–600 nm region (Table 6). For PMP+, the larger errors are located at 540–580 and 710–800 nm. Although we have considered the non-pigment and Ants absorption in this implementation of PMP+, the modelling capability in these two spectral regions is not significantly improved compared with PMP [22]. It is possible that further improvements may require a more accurate determination of leaf average refractive index using a complex refractive index [25].

#### 5.2.3. Pigment Concentration Retrieval Performance

Figure 5 illustrates the pigment retrieval capabilities of PMP+, PD, and P5. The results demonstrate that PROSPECT-MP+ can retrieve not only Chls and Cars concentrations from in vivo leaf DHR and DHT (as do PROSPECT-D and PROSPECT-5), but also the photo-protective pigment (Ants), as does PROSPECT-D, and the subdivisible photosynthetic pigments (Chla and Chlb). PROSPECT-D cannot retrieve leaf Chla and Chlb concentrations from leaf spectra, and PROSPECT-5 cannot do for Chla, Chlb, and Ants information (those are marked with “vacancy” in Figure 4). Moreover, the scatter points with a higher R^2^ (0.3651) from Figure 5d are closer to the 1:1 line than those from Figure 5g with a lower R^2^ (0.2949), which demonstrates that PROSPECT-MP+ can improve the capability of leaf Cars concentration retrieval compared with PROSPECT-5. Regarding Chls and Ants retrieval capabilities, PMP+ is slightly better than PD or P5 depending on their R^2^ values.

The retrieving ability for Chls, Cars, and Ants concentrations from the leaf spectrum is similar for PROSPECT-MP+ and PROSPECT-D. Comparing with PROSPECT-D, although PROSPECT-MP+ has improved the physical feature of the Cars absorption coefficient, there is a significantly difference in the leaf averaged refractive index between the two PROSPECT versions. This could be reason that the retrieval ability of leaf pigments in the PD version is not much improved, and there is a need to explore the relationship between leaf pigment absorption coefficients and leaf averaged refractive index in the future.

#### 5.2.4. Pigment Concentration Retrieval Evaluation

Compared with P5, PMP+, in a similar fashion to PD, can substantially improve Chls and Cars retrieval, as evidenced by the RMSE, SEC, and CV metrics (Table 8). PMP+ can also accurately retrieve Chla and Chlb concentrations from leaf spectra, as shown in other reports [22]. Table 7 also shows that PMP+ is particularly effective for retrieving Ants concentrations, which is similar to the PD version. It is also worth noting that P5 improved the Chls and Cars concentration retrieval performance in comparison with the reports of P5 in the LOPEX93 dataset [22], which may indicate that the measurement of photosynthetic pigments with HPLC (in LOPEX_ZJU) can improve the capabilities of PROSPECT-5.

## 6. Conclusions 

This paper demonstrates that PROSPECT-MP+, which is an extended version of the PROSPECT model in the 400–800 nm region, can produce a new set of multiple photosynthetic and photo-protective pigment absorption coefficients using the LOPEX_ZJU dataset. The determined pigment absorption coefficients (for Chla, Chlb, Cars, and Ants) also possess three key features: (1) they are consistent with the physical principles of pigment absorption spectra; (2) they account for the spectral displacement of absorption peaks within media of different polarities; (3) they quantify the main absorption characteristics of each pigment with the RAF parameter.

To provide some context, the capabilities of leaf spectral modelling and inversion for PROSPECT-MP+ were compared with those of PROSPECT-D and PROSPECT-5. The results were encouraging in that (1) PROSPECT-MP+, like PROSPECT-D, can improve the simulation capabilities of leaf spectra, especially in leaf with Ants present; (2) PROSPECT-MP+ can be used to retrieve leaf Chls, Cars, and Ants with similar accuracies to those of PROSPECT-D, but it improves the accuracy of Cars retrieval compared with PROSPECT-5; (3) PROSPECT-MP+ provides a capability for reliably retrieving individual Chla and Chlb concentrations, like PROSPECT-MP, which is done for PROSPECT-D and PROSPECT-5; (4) PROSPECT-MP+ can also provide a means of accurately retrieving photo-protective pigment (Ants) concentrations from fresh leaf spectra, which is similar to PROSPECT-D.

Our ongoing work is now focusing on improving the description within PROSPECT-MP+ of the optical properties, with explicit parameterizations of the relationship between leaf pigment absorption coefficient and leaf averaged refractive index, leaf surface roughness and surface refractive index, and their interactions with illumination and viewing geometry. Therefore, these future developments of PROSPECT-MP+ should improve the robustness and transferability in the capabilities for retrieval of multiple pigment concentrations, and a synthesis of PROSPECT-MP+ with a canopy RT model will offer opportunities for developing the professional sensor for plant leaf and canopy pigment determining and monitoring.

## Figures and Tables

**Figure 1 sensors-22-03025-f001:**
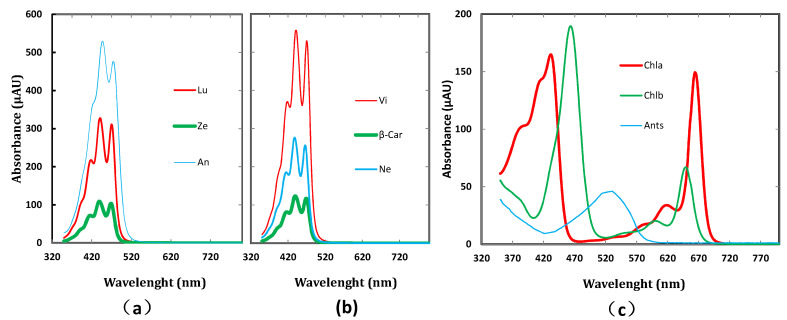
The absorption spectra of pure pigments in acetonitrile/methanol/dichloromethane (from Zhang [32]). The content of Lu, An, Ze in (**a**) and Ne, Vi, β-Car in (**b**) were both 0.2 mg/mL and Chl*a*, Chl*b* in (**c**) were 0.01 mg/mL and Ants in (**c**) were 0.05 mg/mL.

**Figure 2 sensors-22-03025-f002:**
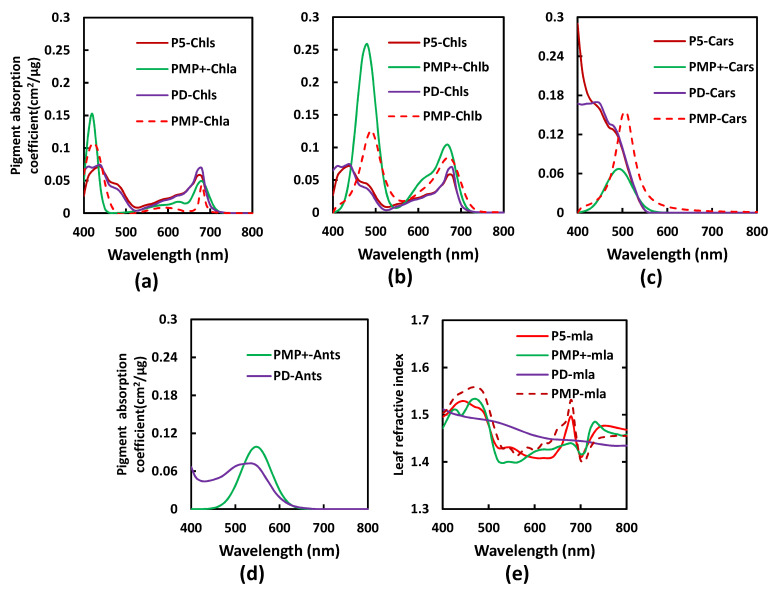
The spectral characteristics of the determined PROSPECT-5 (P5), PROSPECT-D (PD), PROSPECT-MP (PMP), and PROSPECT-MP+ (PMP+) parameters in in vivo leaf: (**a**) Chla-specific absorption coefficient ($K_{\mathrm{Chla}}$); (**b**) Chlb-specific absorption coefficient ($K_{\mathrm{Chlb}}$); (**c**) Cars-specific absorption coefficient ($K_{\mathrm{Cars}}$); (**d**) Ants-specific absorption coefficient ($K_{\mathrm{Ants}}$); (**e**) leaf average refractive index (${\bar{m}}_{\mathrm{la}}$).

**Figure 3 sensors-22-03025-f003:**
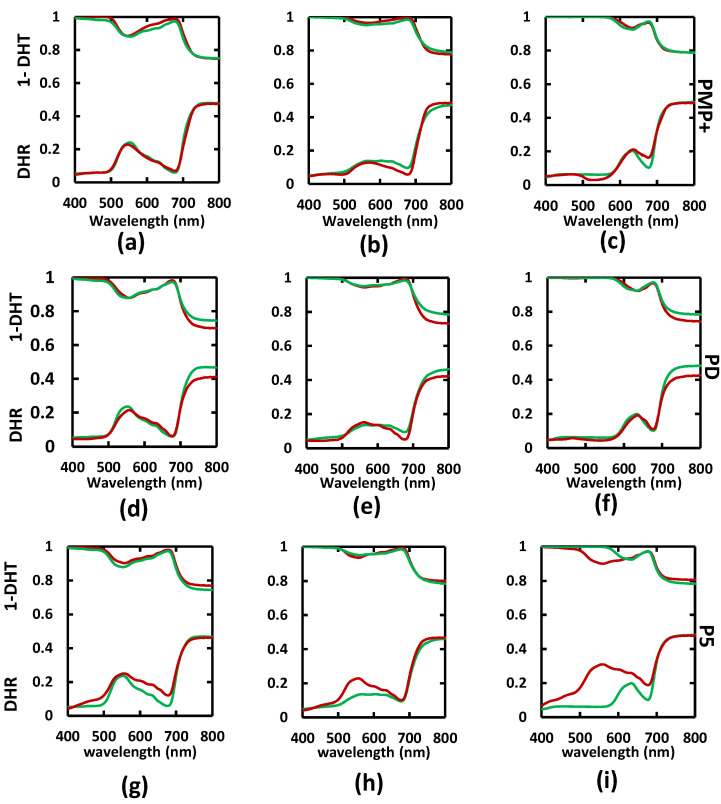
Comparison of measured (green) and simulated (red) reflectance and transmittance spectra for the leaves with different Ants concentrations from PROSPECT-MP+ (PMP+), PROSPECT-D (PD), and PROSPECT-5 (P5), in which (**a**,**d**,**g**) are from the low (0.2I47 μg/cm^2^); (**b**,**e**,**h**) are from the medium (9.8321 μg/cm^2^); and (**c**,**f**,**i**) are from the high (22.5717 μg/cm^2^) concentrations.

**Figure 4 sensors-22-03025-f004:**
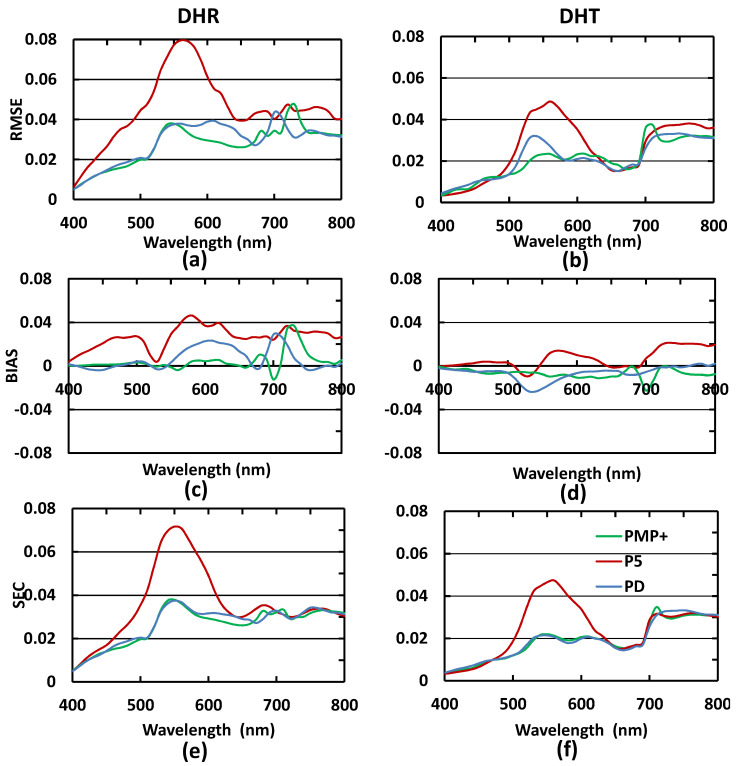
Simulated DHR and DHT spectra from PROSPECT-MP (PMP+) (green line; *n* = 28), PRIOSPECT-D (PD) (blue line; *n* = 28), and PROSPECT-5 (P5) (red line; *n* = 28); (**a**,**c**,**e**) are for the evaluation metrics RMSE, BISA, and SEC of the DHR modelling; (**b**,**d**,**f**) are for the evaluation metrics RMSE, BISA, and SEC of the DHT modelling.

**Figure 5 sensors-22-03025-f005:**
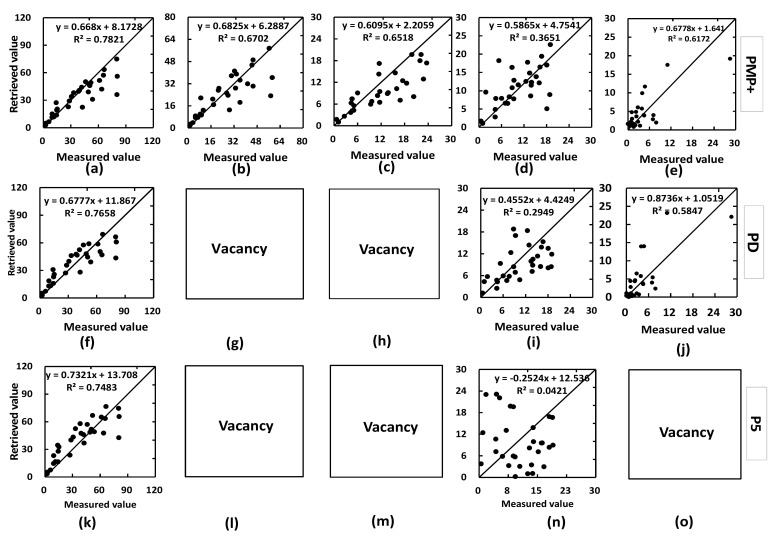
Comparison between measured and retrieved pigment concentrations (µg/cm^2^; *n* = 28) from PROSPECT-MP+ (PMP+), PROSPECT-D (PD), and PROSPECT-5 (P5); (**a**,**f**,**k**) are for Chls concentration; (**b**,**g**,**l**) are for Chla; (**c**,**h**,**m**) are for Chlb; (**d**,**i**,**n**) are for Cars, and (**e**,**j**,**o**) are for Ants. The “vacancy” is expressed for the non-retrieving leaf pigment concentration in the corresponding PROSPECT version.

**Table 1 sensors-22-03025-t001:** Notation.

Symbol	Quantity	Unit	Symbol	Quantity	Unit
** λ **	Wavelength	Nanometer (nm)	$K_{i,j,h}$	Wavelet spectra peek height	cm^2^ μg^−1^ nm^−1^
** N **	Leaf structure parameter	None	Chla	Chlorophyll a	None
** τ **	Leaf transmission coefficient	None	Chlb	Chlorophyll b	None
** $m_{l,a}$ **	Leaf refractive index	None	Cars	Carotenoids	None
** *i* **	Leaf pigment type	None	β-Car	β-carotenoid	None
** *j* **	Absorption peak number	None	Vi	Violaxanthin	None
** $K_{i}\lambda$ **	Pigment absorption coefficients	cm^2^ μg^−^^1^ nm^−^^1^	An	Antheraxanthin	None
** $K_{i,j}\lambda$ **	Leaf pigment absorption peak	cm^2^ μg^−^^1^ nm^−^^1^	Ze	Zeaxanthin,	None
** $K_{i,j,v}$ **	Gauss ratio	None	Ne	Neoxanthin	None
** $A_{i,j,p}$ **	Wavelet spectra peek position	cm^2^ μg^−1^ nm^−1^	Lu	Lutein	None
** $K_{i,j,\Delta \lambda }$ **	Spectral displance	Nanometer (nm)	Ants	Anthocyanins	None

**Table 2 sensors-22-03025-t002:** Description of leaf samples in the LOPEX_ZJU data.

Species No.	Common Name	Species Name	No. of Leaves	Leaf Life Cycle Stage	SPAD Range
1	Loropetalum	*Loropetalum chinense rubrum Yieh*	5	Y, M	22.3–60.5
2	Japan Arrow wood	*Viburnum awabuki*	5	Y, M	32.6–70.2
3	Ginkgo	*Ginkgo*	5	M, S	3.8–41.8
4	Sweet-scented osmanthus	*Osmanthus fragrans*	5	Y, M	15.1–51.5
5	Mulberry	*Morus alba*	4	Y, M	13.7–52.5
6	Moso Bamboo	*Phyllostachysheterocycla*	4	Y, M, S	12.3–52.3
7	Decipiens	*Elaeocarpussylvestris Poir*	5	M, S	1.5–61.0
8	Pterostyrax	*Pterostyrax corymbosus*	5	Y, M, S	4.3–44.0
9	Sapindus	*Sapindusmukurossi*	5	M, S	0.0–42.9
10	Sugar Maple	*Acer saccharum*	5	M, S	0.0–30.3
11	Camphor Tree	*CinnamomumcamphoraPresl.*	5	M, S	4.2–34.7
12	Tea Tree	*Camellia Sinensis*	6	Y, M	34.1–80.4

Note that the notations Y, M and S stand for young leaf, mature leaf and senescence leaf, respectively.

**Table 3 sensors-22-03025-t003:** Leaf biochemical and biophysical measurements ^A^ for the LOPEX_ZJU dataset and range of the ratio between leaf photosynthetic pigments.

Leaf Pigment	Maximum	Minimum	Average	Unit	Chla/Cx ^C^
Chla	94.53	0.04	24.63	μg/cm^2^	1
Chlb	47.49	0.05	12.75	μg/cm^2^	0.4–1.09
Ants	47.22	0.01	4.12	μg/cm^2^	▬
Cars ^B^	44.55	0.24	16.09	μg/cm^2^	0.19–7.04
Lu	17.71	0.02	4.76	μg/cm^2^	▬
An	1.83	0	0.37	μg/cm^2^	▬
Ze	6.99	0.02	1.06	μg/cm^2^	▬
Vi	4.1	0	0.95	μg/cm^2^	▬
Ne	7.43	0	1.85	μg/cm^2^	▬
β-car	15.33	0.02	4.1	μg/cm^2^	▬
Water content	73.83	11.61	52.34	%	▬

Note that A expresses that pigments and water content are provided for the fresh leaves; B expresses the Cars concentration as the sum of Lu, An, Ze, Vi, Ne, and β-Car concentrations in the corresponding leaf samples; C expresses the ratio range of different pigment concentrations between leaf samples **▬** expresses the no considering of the ratio between different pigments.

**Table 4 sensors-22-03025-t004:** The number and position of absorption peak for pure pigment in the 400–800 nm region from a mixed organic solution (modified from Zhang et al. [22,32]).

**Absorption Peak No.**	$A_{\mathrm{Chla},j,p}$ (nm)	$A_{\mathrm{Chlb},j,p}$ (nm)	$A_{\mathrm{Cars},j,p}$ (nm)	$A_{\mathrm{Ants},j,p}$ (nm)
j=1	432	458	418	530
j=2	580	602	443	▬
j=3	618	650	470	▬
j=4	664	▬	▬	▬

Note that **▬** expresses the no absorption peak in the pigment.

**Table 5 sensors-22-03025-t005:** Implementations of PROSPECT-MP+ (PMP+), PROSPECT-5 (P5), and PROSPECT-D(PD) using the LOPEX_ZJU dataset for spectral modelling and pigment retrieval by model inversion. $R_{\mathrm{mea}}$, $T_{\mathrm{mea}}$, and $C_{\mathrm{mea},i}$ stand for the measured leaf DHR, DHT, and pigment concentration and $R_{\mathrm{mod}}$, $T_{\mathrm{mod}}$, and $C_{\mathrm{inv},i}$ for the modeled or retrieved values.

Versions	Dataset	Application	Sample	Input Variables	Algorithm	Output Variable	Description
PMP^+^	LOPEX_ZJU	Forward spectral modelling	28	$C_{\mathrm{mea},i},\;K_{i},K_{0},\;{\bar{m}}_{\mathrm{la}},\;N$	Direct computing for each leaf sample	$R_{\mathrm{mod}},\;T_{\mathrm{mod}}$	$i\;\mathrm{can}\;\mathrm{be}\;\mathrm{Chla},\;\mathrm{Chlb},\;\mathrm{Cars}\;\mathrm{or}\;\mathrm{Ants},\;K_{i},\;{\bar{m}}_{\mathrm{la}}$ from PMP^+^.
PMP^+^	LOPEX_ZJU	Inversion for pigment retrieval	28	$R_{\mathrm{mea}},\;T_{\mathrm{mea}},\;K_{i},K_{0},\;{\bar{m}}_{\mathrm{la}}\;\mathrm{and}\;N$	Minimizing the merit function & a least squares optimization	$C_{\mathrm{inv},i}$	$i\;\mathrm{can}\;\mathrm{be}\;\mathrm{Chla},\;\mathrm{Chlb},\;\mathrm{Cars}\;\mathrm{or}\;\mathrm{Ants},\;K_{i},\;{\bar{m}}_{\mathrm{la}}$ from PMP^+^.
PD	LOPEX_ZJU	Forward spectral modelling	28	$C_{\mathrm{mea},i},\;K_{i},\;{\bar{m}}_{\mathrm{la}}\;\mathrm{and}\;N$	Direct computing for each leaf sample	$R_{\mathrm{mod}},\;T_{\mathrm{mod}}$	i$\;\mathrm{can}\;\mathrm{be}\;\mathrm{Chls},\;\mathrm{Cars}\;\mathrm{or}\;\mathrm{Ants},\;K_{i},\;{\bar{m}}_{\mathrm{la}}$ from PD.
PD	LOPEX_ZJU	Inversion for pigment retrieval	28	$R_{\mathrm{mea}},\;T_{\mathrm{mea}},\;K_{i},\;{\bar{m}}_{\mathrm{la}}\;\mathrm{and}\;N$	Minimizing the merit function & a least squares	$C_{\mathrm{inv},i}$	$i\;\mathrm{can}\;\mathrm{be}\;\mathrm{Chls},\;\mathrm{Cars}\;\mathrm{or}\;\mathrm{Ants},\;K_{i},\;{\bar{m}}_{\mathrm{la}}$ from PD.
P5	LOPEX_ZJU	Forward spectral modelling	28	$C_{\mathrm{mea},i},\;K_{i},K_{0},\;{\bar{m}}_{\mathrm{la}}\;\mathrm{and}\;N$	Direct computing for each leaf sample	$R_{\mathrm{mod}},\;T_{\mathrm{mod}}$	$i\;\mathrm{can}\;\mathrm{be}\;\mathrm{Chls}\;\mathrm{or}\;\mathrm{Cars},\;K_{i},\;{\bar{m}}_{\mathrm{la}}$ from P5.
P5	LOPEX_ZJU	Inversion for pigment retrieval	28	$R_{\mathrm{mea}},\;T_{\mathrm{mea}},\;K_{i},K_{0},\;{\bar{m}}_{\mathrm{la}}\;\mathrm{and}\;N$	Minimizing the merit function & a least squares	$C_{\mathrm{inv},i}$	$i\;\mathrm{can}\;\mathrm{be}\;\mathrm{Chls}\;\mathrm{or}\;\mathrm{Cars},\;K_{i},\;{\bar{m}}_{\mathrm{la}}$ from P5.

**Table 6 sensors-22-03025-t006:** Absorption peak characteristics determined from the in vivo pigment absorption coefficients within PROSPECT-MP+ (PMP+).

Specific Absorption Coefficient	Absorption Peak	$K_{i,j,v}$	$K_{i,j,h}$ (cm^2^/μg)	$K_{i,j,w}$ (nm)	$K_{i,j,p}$ (nm)	$\Delta \lambda_{i,j}$ (nm)	RAF (nm)
$K_{Chla}$	j=1	0.80	0.153	51	419	−13	400–434
	j=2	1.00	0.016	113	591	11	▬
	j=3	0.78	0.008	182	627	9	▬
	j=4	0.37	0.049	25	679	15	659–699
$K_{Chlb}$	j=1	0.45	0.254	60	468	4	442–495
	j=2	0.75	0.017	42	612	9	▬
	j=3	0.44	0.106	57	661	11	639–683
$K_{Cars}$	j=1	0.5	0.067	56	482	39	447–517
$K_{Ants}$	j=1	0.45	0.099	100	544	14	494–594

Note that the symbol “▬” stands for the negligible values in the RAFs because of the low absorbance values of these features; $K_{i,j,p}\;=\;A_{i,j,p}\;+\;K_{i,j,\Delta \lambda }$.

**Table 7 sensors-22-03025-t007:** Global performance evaluation of simulated leaf spectra DHR (Directional Hemispherical Reflectance) and DHT(Directional Hemispherical Reflectance) from PROSPECT-MP+ (PMP+), PROSPECT-D (PD), and PROSPECT-5 (P5) (*n* = 28).

Spectrum Type	Model Implementation	RMSE	BIAS	SEC
DHR	PMP+	0.027	0.004	0.026
	PD	0.029	0.007	0.027
	P5	0.045	0.011	0.040
DHT	PMP+	0.021	−0.007	0.019
	PD	0.023	−0.001	0.020
	P5	0.027	−0.001	0.027

**Table 8 sensors-22-03025-t008:** The validation of pigment concentration retrievals from in vivo leaf spectra by PROSPECT-MP+ (PMP+), PROSPECT-D (PD), and PROSPECT-5 (P5).

Performance Types	PMP+					PD			P5	
Pigment Types	Chls	Chla	Chlb	Cars	Ants	Chls	Cars	Ants	Chls	Cars
RMSE μg/cm^2^	12.51	11.69	6.54	8.18	3.17	12.56	8.93	3.8	13.70	10.24
BIAS μg/cm^2^	−3.38	−0.16	−3.22	0.76	0.07	−3.44	−2.05	0.26	1.99	5.05
SEC μg/cm^2^	12.04	11.69	5.67	8.15	3.17	12.21	8.47	3.79	13.55	8.23
CV %	27.03	31.84	39.37	39.24	45.42	33.03	43.49	90.24	37.19	70.09

## Data Availability

Not applicable.

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
