# Peer review of "PROSPECT-PMP+: Simultaneous Retrievals of Chlorophyll a and b, Carotenoids and Anthocyanins in the Leaf Optical Properties Model"

_sensors, 2022, doi:10.3390/s22083025_

Round 1

Reviewer 1 Report

The manuscript introduces an improved version of the familiar Prospect model for leaf transmission and reflectance, which has the capability of retrieving photochemical pigments other than chlorophylls. The description of the research problem, its purpose, the methods used, and the results obtained were complete, and it's clear what the overall conclusions of the paper were.  I actually have very few comments about the scientific content of the paper.  Most of my concerns arise from the fact that there are numerous instances where the written parts are difficult to understand. Specific comments follow.

Science Comments.

1. In Section 5.2.1, a validation data set is used. Was this collected separately from the data set used to calibrate the new model? If so, how large was it and was it collected in the same way as the other data?

2. Figure 3 on my review copy was difficult to read and understand. The y-axes were labeled RMSE, BIAS, and SEC. Is this correct? Some of these actually look like reflectance or transmission spectra, which is more consistent with what the caption and text suggest that they are. This is especially true of a, b, e , f. Can you clarify what, exactly, this figure sis showing?

3. In Section 5.2.3, More validation is shown. Again, can you be more specific about the source of the data used?

4. The section from about line 200 to 239 is not very clearly explained. The information in Table 3 was especially hard for me to understand. I looked up absorption spectra for the pigments presented (Chla, Chlb, Cars, Ants) and while some of the peaks indicated in the table matched, others did not. The Zhang papers cited were not all that clarifying, either. The tabulated values in the 2017 paper did not match the ones here. The ones in the 2020 Zhang paper do match these, and can be seen on Figure 2. However, it’s not clear to me why common, published pigment absorption data do not match the ones shown in Zhang’s paper and used here. Is this a consequence of the pigments being in a mixture? If so, this needs to be explained. If I were planning to replicate the process, it seems to me that this would be crucial information. Would it be possible to insert a figure, similar to Figure 2 in the Zhang, 2020 paper, that shows these features? This was the most confusing part of the paper for me, and also (it seems) the part that separates the MP+ model from the MP version.

Comments on Presentation

The manuscript contains numerous instances of misspellings, improper capitalization, wrong word choices, and difficult to understand sentences. I would urge the authors to have the manuscript reviewed for English usage by an experienced and knowledgeable editor. The list below is not complete, but gives examples of the types of errors I an referring to. I understand that writing in a language other than one’s own is inherently difficult, but I the paper would be much more effective with a thorough revision for English usage.

Some specifics examples (not a complete list):

Line 50 “Improved” should not ne capitalized.

Lines 54 “…especially no-considering...” should probably be not considering.

Line 57 depend is the wrong tense. Should be depends

Line 79 monitor is the wrong tense. Should be monitoring.

Lines 176-177 Do you mean “Simultaneous separation of leaf multiple photosynthetic pigments...” here?

Lines 179-181 This sentence really doesn’t make sense.

Lines 197-199 Ialso can’t really determine what is meant here.

Line 255 algorithm is misspelled.

Line 290 Is “environmental polarity” really correct here?

I stopped flagging usage errors at this point. Again, I would strongly urge a thorough review of the paper for correct English. This is a good piece of research, and it needs to be understood by its readers.

Author Response

Dear Reviewer:

Thank you very much for providing valuable comments and suggestions. Below are our point-to-point responses. In the revised manuscript file, Language had been revised throughout the manuscript by a professional. 

please see the attachment (authou-coverltter-182119589.v1) for our response.

Reviewer 2 Report

It is an interesting study on improving a model for estimating the concentration of assimilating pigments with a focus on carotenoids and anthocyanins. Specific comments:
1. In table 1 the name of the species is not written correctly, please check.
2. In Table 2 please specify what Lu, An, Ze, Vi, and Ne are.
L 133 What does "Agelent" represent? Please check everywhere in the text.
Please include a list at the beginning of the article or at the end with all the abbreviations used.

Author Response

Dear Reviewer:

Thank you very much for providing valuable comments and suggestions. Below are our point-to-point responses.

Point 1. In table 1 the name of the species is not written correctly, please check.

Response: We had checked the name of these species throughout the table, and correct the errors (see Table 2 in the revised manuscript)

Point 2. In Table 2 please specify what Lu, An, Ze, Vi, and Ne are.

Response: The abbreviations had listed and detail described in Table 1.

Point 3. L133 What does "Agelent" represent? Please check everywhere in the text.

Response: This error had been modified into Agilent in the revised manuscript file.

Point 4. Please include a list at the beginning of the article or at the end with all the abbreviations used.

Response: The abbreviations had listed and detail described in Table 1.

Round 2

Reviewer 2 Report

The authors made all the changes. I believe that the article can be published in this form.